# Effects of Biochar and Nitrogen Application on Rice Biomass Saccharification, Bioethanol Yield and Cell Wall Polymers Features

**DOI:** 10.3390/ijms232113635

**Published:** 2022-11-07

**Authors:** Izhar Ali, Muhammad Adnan, Anas Iqbal, Saif Ullah, Muhammad Rafiullah Khan, Pengli Yuan, Hua Zhang, Jamal Nasar, Minghua Gu, Ligeng Jiang

**Affiliations:** 1College of Agriculture, Guangxi University, Nanning 530004, China; 2Department of Food Engineering, Pak-Austria Fachhochschule, Institute of Applied Sciences and Technology, Khanpur Road, Mang, Haripur 22620, Pakistan

**Keywords:** rice, biomass digestibility, cell wall features, biochar, bioethanol, nitrogen

## Abstract

Rice is a major food crop that produces abundant biomass wastes for biofuels. To improve rice biomass and yield, nitrogen (N) fertilizer is excessively used, which is not eco-friendly. Alternatively, biochar (B) application is favored to improve rice biomass and yield under low chemical fertilizers. To minimize the reliance on N fertilizer, we applied four B levels (0, 10, 20, and 30 t B ha^−1^) combined with two N rates (low-135 and high-180 kg ha^−1^) to improve biomass yield. Results showed that compared to control, the combined B at 20–30 t ha^−1^ with low N application significantly improved plant dry matter and arabinose (Ara%), while decreasing cellulose crystallinity (Crl), degree of polymerization (DP), and the ratio of xylose/arabinose (Xyl/Ara), resulting in high hexoses (% cellulose) and bioethanol yield (% dry matter). We concluded that B coupled with N can alter cell wall polymer features in paddy rice resulting in high biomass saccharification and bioethanol production.

## 1. Introduction

As the consumption of fossil fuels rises and the energy problem worsens, future generations will be put under strain to meet the ever-increasing need for food and energy in the contemporary period. New energy production methods are needed to reduce these problems and reduce environmental risks. Several alternative energy production technologies have been created, with crops biomass energy generation receiving particular attention due to its ecologically beneficial nature [1,2]. Rice is the most cultivated crop in Guangxi province of China and a staple food worldwide, resulting annually in more than 800 million metric tons of biomass [3,4]. However, a huge quantity of agricultural biomass is burned in the field and buried in soil in southern China, which is not environmentally friendly since it harms soil fauna and degrades soil quality. Moreover, fertilizer management significantly affects rice biomass and its yield [5,6,7], and to improve rice yield and biomass, farmers excessively use chemical fertilizer, especially N, which is not eco-friendly [8,9]. To cope with this problem, many studies reported that biochar has the potential to decrease the requirement of N fertilizer without compromising rice yield and biomass [3,9,10,11].

Biochar is a carbon-rich, refractory organic substance made by the controlled combustion of organic material (biomass) from agricultural and forestry wastes, animal bones, algae, and animal manures [3,12]. The application of biochar not only improves physiochemical soil properties [9,13] but also reduces greenhouse gas emissions [14,15,16]. Previously, it was reported that biochar can improve rice biomass digestibility [3]. However, the impact of biochar in combination with nitrogen fertilizer on rice crop cellulose, hemicelluloses, lignin, hexoses, and bioethanol yield has not yet been reported.

Generally, cellulose, hemicellulose polysaccharides, and lignin are the main components of lignocellulose biomass [17]. Cellulose is made of a β-1, 4 bonded glucan chain, formed by microfibrils and bound together by hydrogen bonds. These hydrogen bonds are the key variables that adversely affect biomass digestibility and provide stability, and they play a critical role in cellulose crystallinity [18,19,20]. The lignin composition includes complex aromatic polymers that are cross-linked to hemicellulosic polysaccharides. These complex polymers provide an impermeable matrix that gives mechanical strength to the cell wall of the plant and protects the cell wall of polysaccharides against various chemicals and enzymatic hydrolysis [21,22]. Hemicellulose is a type of heterogeneous polysaccharide that improves biomass saccharification by lowering the crystallinity of lignocelluloses [23,24]. Many other factors influence the ability of cellulose to break down during enzyme hydrolysis, including coarse particles, more crystalline form, low porosity, less usable surface area, etc. [25,26,27,28,29,30,31]. Thus, regarding the importance of biomass composition, it is important to determine the impacts of different biochar rates in combination with N fertilizer on cell wall features and cellulose monomers of rice crops.

In the present study, we collected the rice crop biomass samples from the experimental field treated with different biochar and N fertilizers. Then, we determined major alterations of cell wall compositions and wall polymer features in the mature straw of rice crops. Based on comparative and correlative analyses, this study identified the main factors of plant cell walls that predominately determine biomass enzymatic saccharification and bioethanol production in rice crops under different treatments.

## 2. Results 

### 2.1. Biochar and Nitrogen Impact on Rice Plant Dry Matter

Results showed that all biochar-applied treatments resulted in higher plant dry weight than non-biochar-treated plots (Figure 1). Compared to sole N applied treatment(N1B0), the combined treatments N1B2, N1B3, N2B2, and N2B3 increased plant dry weight by 9%, 15%, 14%, and 16%, respectively, during 2019. Similarly, during 2020, an increase of 6%, 8%, and 9% were recorded in N1B3, N2B2, and N2B3 compared to control (N1B0). 

### 2.2. Biochar and Nitrogen Impact on Cell Wall Compositions of Biomass Residues

Rice plants harvested from sole urea applied treatments had the lowest cellulose, hemicelluloses, lignin, pectin, and starch contents compared to all biochar rates applied treatments. Biochar in combination with nitrogen fertilizer significantly affected cell wall composition of biomass residues i.e., cellulose, hemicelluloses, lignin, pectin, and starch content in rice straw (Table 1). Compared with control, combined biochar treatments with nitrogen fertilizer slightly improved cellulose content ranging up to 33% during both years. Similarly, hemicelluloses, lignin, pectin, and starch content were improved in combined biochar treatments with nitrogen fertilizer ranging up to 21.9%, 17%, 2.7%, and 15.2%, respectively, compared to sole N applied treatments. 

### 2.3. Biomass Saccharification under 1% NaOH with and without Tween-80

Biomass digestibility refers to the hexose yield (% cell cellulose) or total sugar (% dry matter) released during enzymatic hydrolysis from the pretreated biomass residues. In the current study, we used a mild alkali pretreatment (1% NaOH) without Tween-80, and with a co-supply of 1% Tween-80 on rice crops harvested from different biochar and nitrogen fertilizer applied treatments. Biochar, in combination with nitrogen fertilizer, significantly affected hexose yields (with and without Tween-80) during both years (Figure 2). Analysis of the results showed that biochar at the rate of 20 to 30 t ha^−1^ (N1B2, N1B3, N2B2, and N2B3) significantly improved hexoses yield (with Tween-80) by 8–15% (% cellulose) as compared to sole urea applied treatments during both years. Similarly, compared to control without Tween-80, hexose yield was higher by 18–21% in combined biochar and nitrogen fertilizer treatments. Notably, Tween-80 significantly enhanced the hexose yield in all nitrogen coupled with biochar supply. 

### 2.4. Bio-Ethanol Yield (% Dry Matter) from Yeast Fermentation

Classic yeast fermentation was performed for bioethanol production using total soluble sugars released from 1% NaOH pretreatment and sequential enzymatic hydrolyses in representative samples (Figure 3). Biochar, in combination with nitrogen fertilizers, significantly improved bioethanol yield (% dry matter) with and without Tween-80. Compared with control, ethanol yield with and without Tween-80 of 13.5–14.6% and 16.5–17.8% were recorded in biochar-applied treatments, respectively. Furthermore, during 2019, bioethanol yield was higher by 24–25% with Tween-80 in biochar-applied treatments compared to non-biochar treatments. 

### 2.5. Changes in Cellulose Features and Hemicelluloses Monomers 

To understand why rice crops showed high biomass saccharification and bioethanol production in biochar combined with nitrogen fertilizer treatments, we compared cell wall polymer features affecting biomass digestibility with biochar and nitrogen applications (Figure 4). Cellulose Crl and DP are the main inhibitors of biomass enzymatic saccharification [4]. Surprisingly, in this study, biochar application combined with low nitrogen fertilizer increased cellulose content and significantly reduced cellulose Crl and DP by 6 to 10% as compared to non-biochar treatments during both years (Figure 4A,B). Furthermore, Ara was increased by 25–27%, while Xyl/Ara was reduced by 15–20% during both years with biochar application. 

### 2.6. Mechanism of Biochar and Nitrogen Application Improvements on Biomass Digestibility

To determine the effect of biochar and nitrogen fertilizer on biomass digestibility, a correlation analysis was carried out among biochar plus nitrogen fertilizers and cell wall components. The results showed that the application of biochar combined with nitrogen fertilizer significantly increased plant dry matter, cellulose, and hemicellulose contents (Figure 5A). Moreover, a significant positive correlation between hexose (% cellulose) and Ara% was recorded, while a negative relationship was observed for cellulose Crl, Dp, and Xyl/Ara with biochar and nitrogen fertilizers (Figure 5B). Based on these findings, a hypothetical mechanism of biochar and nitrogen fertilizer was proposed affecting biomass digestibility in rice. First, biochar combined with N fertilizer treatments could consistently improve plant dry matter production in rice (Figure 1). Second, the combined treatments of biochar and nitrogen fertilizer reduced cellulose Crl and DP value and Xyl/Ara ratio (Figure 4); finally, the increased biomass enzymatic digestibility under various biochar and nitrogen applied treatments may be explained by decreased cellulose Crl, DP, and hemicellulose Xyl/Ara as a critical negative effect. Overall, the model showed that cell wall characteristics (cellulose, DP, and Xyl/Ara) could influence biomass digestibility in a favorable way by lowering cellulose CrI in rice crops treated with biochar and nitrogen fertilizer. Additionally, it will be interesting to investigate in the future whether biochar and N fertilizer similarly affect plant resistance to biotic and abiotic stresses, given that they may noticeably alter wall polymer characteristics.

## 3. Discussion

Rice dry matter accumulation is positively associated with biomass enzymatic saccharification, including hexoses, pentoses, and total sugars [3]. In the present study, dry matter production resulted higher in biochar treatments as compared to individual urea-applied plots. The possible reason for these increments might be due to biochar amendment releasing nutrients slowly throughout the growing season, which can fulfill the plant N requirement [10,32]. Biochar, in combination with nitrogen fertilizer, significantly improved dry matter production of rice compared to sole urea treatments [9,11]. Previously, it was reported that biochar improves soil physiochemical properties [33,34,35], enhances photosynthesis rate [36], and improves root morphological attributes [11], which can consequently increase plant dry weight. Moreover, biochar also improves nitrogen accumulation in plants [37,38], resulting improvement in rice biomass production. Based on the results discussed above, we observed that the addition of biochar to soil with nitrogen fertilizer improves N availability slowly throughout the rice growing season, resulting improving rice biomass.

Cell wall compositions, including cellulose, hemicellulose polysaccharides, and lignin, are the main components of lignocellulose biomass [17,39]. In the current experiment, cellulose, hemicelluloses, lignin, pectin, and starch content were enhanced in biochar-applied treatments, while sole urea-applied treatments had the lowest cellulose, hemicelluloses, lignin, pectin, and starch contents. Therefore, this study indicated that higher biomass yield may be due to biochar structure, macro and micro nutrient contents, which considerably improved rice dry matter accumulation [9]. Furthermore, similar results were reported by Ali et al. [3] that biochar plus nitrogen significantly improved rice straw biomass enzymatic saccharification as compared to control. Zahoor et al. [4] reported that N application could increase cellulose content from 8–23% in rice cultivars. 

Tween-80 is a non-ionic surfactant that has been widely used to enhance the enzymatic digestibility of pretreated biomass, having high stability, good compatibility, and high solubility in the aqueous enzymatic hydrolysis system [4,40]. Our results demonstrated that biochar fertilizer significantly affected hexoses and bioethanol yields (with and without Tween-80). The addition of Tween-80 significantly enhanced biomass digestibility in miscathus, reed, and wheat [4,40,41]. In this study, the addition of 1% Tween-80 significantly enhanced the hexose yield in all nitrogen coupled with biochar supply. The possible explanation for these increments is that the application of biochar and nitrogen fertilizer can improve soil fertility and rice dry matter production [9,10], resulting in improvements in rice biomass enzymatic saccharification under alkali pretreatments [3]. Another possible reason is that biochar improved plant dry matter and N uptake [37,42], and Zahoor et al. [4] reported that an improvement in the supply of N fertilizer could improve biomass enzymatic saccharification in rice. The lowest bioethanol yield was recorded in control treatments. Our results are consistent with Ali et al. [3] and Zahoor et al. [4] who found that biochar and nitrogen fertilizer improved biomass digestibility and bioethanol yield compared to control, respectively. 

Cellulose features and hemicellulose monomers, such as Crl, DP, Ara, and Xyl/Ara, are the main inhibitors of biomass enzymatic saccharification [43,44]. In the present study, Crl, Xyl/Ara, and DP were reduced and Ara% was increased in combined treatments of biochar and nitrogen fertilizers. High Ara% and low Xyl/Ara positively affect biomass digestibility in agriculture biomass residues [40,45]. The results demonstrate that biochar coupled with low nitrogen fertilizer significantly increased plant dry matter and altered the cell wall features resulting in high biomass digestibility and bioethanol production. 

In the correlation analysis, we observed that hexoses released (% cellulose) were negatively correlated with Crl, Dp, and Xyl/Ara while positively correlated with DM and ARA. A similar relationship of cell wall polymers with biomass digestibility was reported by Zahoor et al. [4] under different nitrogen-applied treatments. In addition, Ali et al. [3] reported that biochar can increase rice dry matter production and biomass digestibility under different enzymatic hydrolysis. Furthermore, the mechanism showed that biochar and nitrogen fertilizer improved DM, and decreased Crl, Dp, and Xyl/Ara, resulting increase in Ara%, which consequently improved biomass digestibility. 

## 4. Material and Methods 

### 4.1. Experimental Location and Design 

A field experiment was performed in 2019 and 2020 at GXU, China, and it is categorized as a subtropical monsoon climate having annual precipitation of 1080 mm with an average minimum temperature of 23.4 °C and an average maximum temperature of 32.5 °C. The soil of the experimental field was ultisol, and the basic properties were: pH (5.94), available phosphorous 23.1 mg kg^−1^, soil organic matter 25.8 g kg^−1^, available N 134.7 mg kg^−1^, total N 1.35 g kg^−1^, soil organic carbon15.10 g kg^−1^, and with 233.6 mg kg^−1^of available potassium.

The experimental design was a randomized complete block design (RCBD) with three replicates with each triplicate plot size of 3.9 × 6 m (23 m^−2^). Biochar amendment was used with four different levels, B0, B10, B20, and B30 ton ha^−1^, while two levels of N (N1-135 and N2-180 kg ha^−1^) were applied. The treatments were combined, and the noodle rice variety “Zhenguiai” was sown with uniform seedlings. This variety is mostly cultivated variety in southern China for producing noodles. The previously reported optimal phosphorous and potassium doses of 75 and 150 kg ha^−1^ dose were applied, respectively. 

### 4.2. Biochar (Production) and Soil Properties

Local cassava variety straw was used as a raw material to produce biochar in kiln, with temperature ranging from 300 to 500 °C. The cassava straw biochar had a specific surface area of 2.46 m^2^g^−1^, an average pore diameter of 3.37 nm, and a C:N ratio of 124.12. N, P, K, C, H, and S content of biochar was 5.43, 46.33, 48.33, 674.00, 3.81, and 2.39 g kg^−1^, respectively. The basic soil physiochemical properties prior to experiments are described in our previous study [3]. 

### 4.3. Measurement and Analysis

#### 4.3.1. Chemical Pretreatments and Biomass Enzymatic Saccharification

The pretreatment of rice biomass sample was assessed by following the previously described methods of Zahoor et al. [4] and Ali et al. [3], thus, 0.3 g of rice biomass ground sample were incubated with 6 mL of 1% NaOH for 2 h at 50 °C with shaking at 150 rpm. The obtained pellet was washed with distilled water continuously to reach a neutral pH value. The pellet was washed with mixed cellulase buffer to maintain the pH. Mix cellulase enzyme was added at a concentration of 2 g per liter to the rice biomass and was shaken at 150 rpm at 50 °C for 48 h. Furthermore, 1% tween-80 was added to determine its impact on biomass saccharification in rice crops. The obtained supernatant was used for pentose and hexose determination.

#### 4.3.2. Biomass Composition Analysis

Rice biomass composition (soluble sugars, lipid, starch, and pectin) was assessed by the method of wall polymer extraction, a well-established method reported by Wu et al. [46] and Peng et al. [47].

#### 4.3.3. Determination of Hexoses and Pentoses

Hexoses and pentose were assessed by following anthrone and orcinol methods, as previously described by Aftab et al. [48] and Zahoor et al. [4].

### 4.4. Hemicelluloses Monosaccharides and Lignin Determination 

High-performance anion-exchange chromatography (HPAEC) was used to assess the monosaccharide composition of hemicelluloses by following the procedure of Li et al. [49]. According to the National Renewable Energy Laboratory’s analytical procedure, the total lignin was produced using a two-step acid hydrolysis method, as previously described by Wu et al. [46].

### 4.5. Cellulose Crystallinity Detection

The Rigaku-D/MAX (Uitima III, Japan) was used to determine cellulose crystallinity (CrI) using the X-ray diffraction method, as described by Zhang et al. [20]. 

### 4.6. Determination of Cellulose DP

The viscosity method was used to determined the degree of polymerization of cellulose in rice biomass samples, as described by Zahoor et al. [4].

### 4.7. Determination of Bioethanol

Bioethanol assay in rice biomass samples was determined by using the *Saccharomyces cerevisiae* strain, as previously described by Li et al. [49]. 

### 4.8. Statistical Analysis

Statistical analysis among the eight treatments was performed by Statistix 8.1 software. For all evaluated features, correlation analysis was performed using Pearson correlation analysis at the level of significance of (* *p* 0.05 and ** *p* 0.01). The bar graphs and correlation graphs were constructed using GraphPad Prism 5 software. The average values of the original triple samples were used to calculate the measured attributes.

## 5. Conclusions

Biochar, in combination with N fertilizer, significantly increased biomass yield and biomass digestibility of rice crop. Biochemical analysis showed that cellulose DP and Crl values and hemicellulosic, Xyl/Ara monomers ratio were reduced by biochar combined with nitrogen fertilizer which resulting an increase in biomass digestibility. In addition, correlation analysis exhibited that both cellulose DP, Crl, and Xyl/Ara ratio were negatively correlated with Ara, dry matter production and biomass digestibility. Therefore, we concluded that adding 20–30 tons of biochar ha^−1^ to soil can improve rice plant dry matter and biomass digestibility under a lower N rate of 135 kg ha^−1^. Thus, it is suggested that using biochar in combination with a lower N fertilizer rate is an environmentally friendly approach to obtain optimum bioethanol yield from rice and other agricultural crops. 

## Figures and Tables

**Figure 1 ijms-23-13635-f001:**
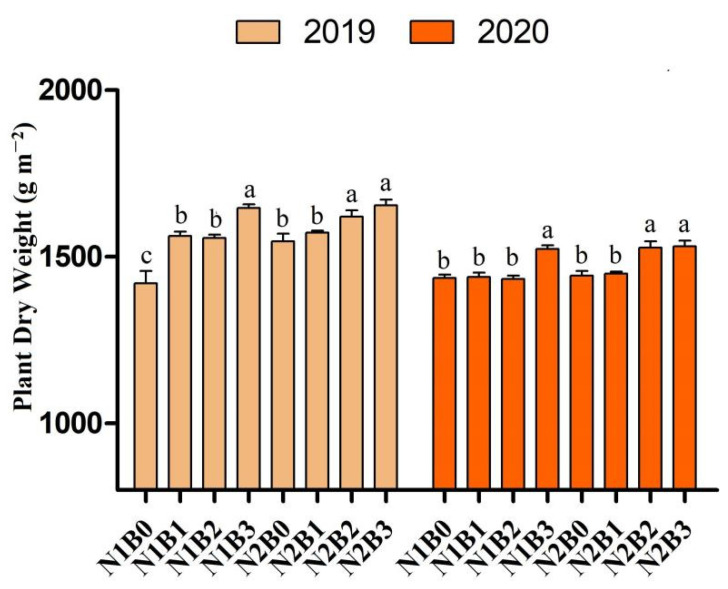
Total biomass production in rice plants under different biochar and nitrogen fertilizer treatments during 2019 and 2020. Different letters on bars are not significantly different at *p* < 0.05.; Bars indicate means ± SD (n = 3).

**Figure 2 ijms-23-13635-f002:**
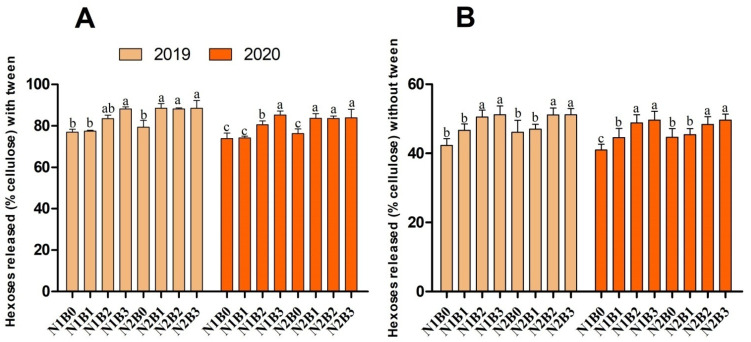
Changes in hexose yields (% cellulose) released with tween (**A**) and without tween (**B**) of rice straw treated with biochar and nitrogen fertilizers. Different letters on bars are not significantly different at *p* < 0.05. (n = 3).

**Figure 3 ijms-23-13635-f003:**
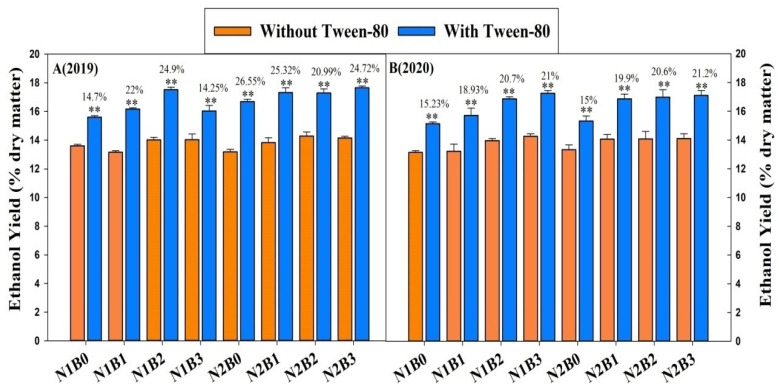
Impact of biochar and nitrogen fertilizer on bioethanol yields from biomass process in rice. ** above the column represent significant difference among the biochar and biochar treatments by *t*-test at *p* < 0.05 or *p* < 0.01 (n = 3). % value indicates the increase in the same treatment with Tween-80 from without Tween-80.

**Figure 4 ijms-23-13635-f004:**
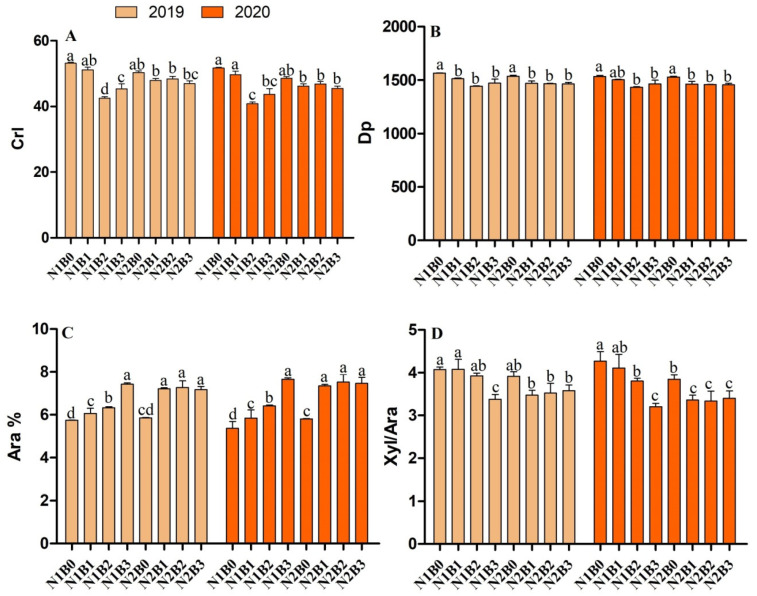
Biochar and nitrogen fertilizer effect on cell wall polymers feature (**A**) crystalline index (CrI) of cellulose, (**B**) degree of polymerization of cellulose, (**C**) arabinose proportion of hemicellulose, (**D**) ratio of xylose and arabinose. Different letters on bars are not significantly different at *p* < 0.05. (n = 3).

**Figure 5 ijms-23-13635-f005:**
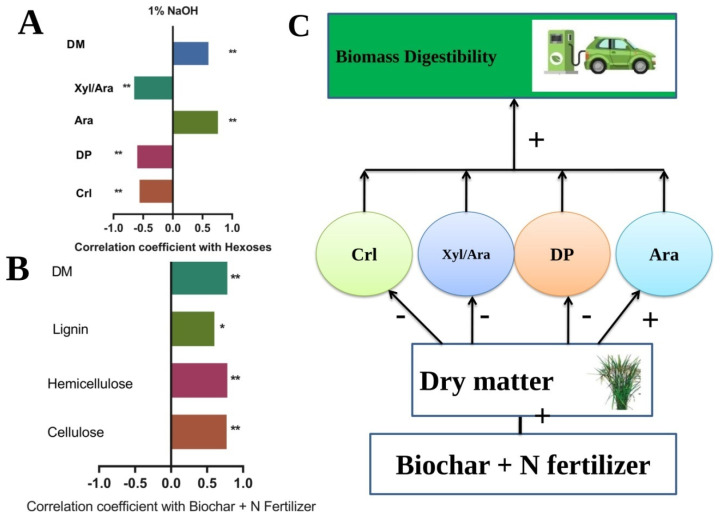
Mechanism involving the primary wall polymer characteristics has an influence on biomass enzymatic saccharification in rice under biochar and nitrogen fertilizer. (**A**) The correlation coefficient between biochar + N fertilizers, cellulose, hemicelluloses, lignin, and DM. (**B**) Correlation analysis between hexoses, Crl, Dp, Ara, Xyl/Ara, and DM. (**C**) A model for improving biomass yield and enzymatic digestibility with biochar and N fertilizer by changing wall polymer characteristics. * and ** indicate a significant coefficient correlation value at *p* < 0.005 and 0.001. (+) and (−) represented increased and reduced effects, respectively.

**Table 1 ijms-23-13635-t001:** Biomass compositions (% dry matter) in raw materials and SE residues of rice treated with biochar and nitrogen fertilizer.

Treatments 2019	Cellulose	Hemicelluloses	Lignin	Pictin	Starch
N1B0	31.22 ± 1.22 b	19.022 ± 1.5 b	16.603 ± 1.0 ab	2.218 ± 0.09 c	14.120 ± 2.3 b
N1B1	32.72 ± 1.73 ab	20.387 ± 1.7 ab	16.166 ± 0.9 b	2.267 ± 0.89 c	14.960 ± 1.6 b
N1B2	33.71 ± 2.58 a	21.640 ± 2.3 a	17.023 ± 1.8 a	2.275 ± 0.26 c	15.277 ± 2.3 a
N1B3	33.30 ± 2.32 a	21.967 ± 2.4 a	17.036 ± 1.8 a	2.440 ± 0.17 b	15.413 ± 1.5 a
N2B0	31.57 ± 3.34 b	19.567 ± 1.3 b	16.182 ± 1.2 ab	2.033 ± 0.88 c	14.362 ± 1.2 b
N2B1	32.43 ± 2.29 ab	20.433 ± 2.2 ab	16.823 ± 2.7 ab	2.433 ± 0.29 b	14.853 ± 1.5 b
N2B2	33.30 ± 2.52 a	21.300 ± 2.7 a	17.290 ± 1.9 a	2.600 ± 0.05 ab	15.241 ± 1.7 a
N2B3	33.27 ± 3.26 a	21.267 ± 2.4 a	17.156 ± 2.8 a	2.733 ± 0.23 a	15.246 ± 1.9 a
**2020**					
N1B0	30.80 ± 2.2 b	18.119 ± 1.6 b	16.129 ± 2.5 b	2.267 ± 0.2 c	14.776 ± 0.9 c
N1B1	32.31 ± 2.3 a	19.517 ± 1.8 ab	16.210 ± 1.8 b	2.237 ± 0.3 c	15.650 ± 1.2 ab
N1B2	33.38 ± 2.5 a	20.640 ± 2.3 a	16.956 ± 1.9 ab	2.277 ± 0.28 c	15.847 ± 1.9 a
N1B3	32.93 ± 1.3 a	20.967 ± 2.6 a	17.254 ± 1.8 a	2.477 ± 0.19 b	15.983 ± 1.6 a
N2B0	31.18 ± 1.2 b	18.567 ± 1.9 b	16.323 ± 1.7 b	2.133 ± 0.35 c	14.932 ± 1.2 c
N2B1	32.11 ± 2.2 a	19.433 ± 1.8 ab	17.056 ± 2.1 a	2.533 ± 0.58 b	15.423 ± 17 b
N2B2	32.95 ± 2.5 a	20.523 ± 2.2 a	17.077 ± 1.9 a	2.700 ± 0.29 ab	15.811 ± 1.5 a
N2B3	32.90 ± 3.6 a	20.933 ± 2.7 a	17.098 ± 1.6 a	2.833 ± 049 a	15.816 ± 1.8 a

Note: Lettering within the column bars represents significant differences between treatments by LSD tests at *p* < 0.05 and 0.01. Data as mean ± SD (n = 3).

## Data Availability

The data and materials will be made available on demand.

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
