# Peer review of "Effects of Biochar and Nitrogen Application on Rice Biomass Saccharification, Bioethanol Yield and Cell Wall Polymers Features"

_ijms, 2022, doi:10.3390/ijms232113635_

Round 1
Reviewer 1 Report
This is a well written article on "Biochar and Nitrogen Application Largely Enhanced Biomass Saccharification and Bioethanol Yield by Altering Cell Wall Polymers Features". Although the research was conducted following a quite solid experimental design.
Here are my recommendations:
- The experiment was conducted using a noodle rice variety. The Authors should explain why this particular variety was selected.
- When describing the cassava row materials utilized as soil amendments the variety utilized should also be analyzed.
- I'd suggest the Authors clarify that study was conducted on rice in the tile and in the introduction.
- Several treatments were applied, however a t-test was conducted. My opinion is that an ANOVA should be conducted and letters instead of asterisks should be utilized in graphs and figures.
- Overall this is an interesting paper but the title, the introduction and the results and discussion are too general for this journal. I'd suggest the Authors to divide the result from the discussion and to rewrite these two paragraphs in order to make it more scientifically sound and more appropriate for a publication in IJMS.
- The conclusions also need to be more specific and needs to relate to the crop used for the study.
Author Response
This is a well written article on "Biochar and Nitrogen Application Largely Enhanced Biomass Saccharification and Bioethanol Yield by Altering Cell Wall Polymers Features". Although the research was conducted following a quite solid experimental design.
Here are my recommendations:
- The experiment was conducted using a noodle rice variety. The Authors should explain why this particular variety was selected.
Response: Thanks for your suggestion, rice noodle variety are mostly used in this province for cultivation, which produced abundant biomass. So we used noodle rice variety. The purpose of its cultivation is to use it as raw material for making noodles. We added the sentence in section 4.1 with a yellow highlights.
- When describing the cassava row materials utilized as soil amendments the variety utilized should also be analyzed.
Response: Thanks for your suggestion; Cassava was a local variety that has no specific name. Cassava was used as raw material for producing biochar. The reason for using cassava straw is that cassava is also mostly cultivated in our province. . We clarify it in section 4.2 with a yellow highlight.
- I'd suggest the Authors clarify that study was conducted on rice in the tile and in the introduction.
Response: Thanks for your suggestion; the title is modified.
- Several treatments were applied; however, a t-test was conducted. My opinion is that an ANOVA should be conducted and letters instead of asterisks should be utilized in graphs and figures.
Response: Thanks for your suggestion; we conducted ANOVA using factorial design, and the figures and tables are modified. However, Figure 3 indicates the difference between the same treatment with and without Tween-80. The % value indicates the increase in the same treatment with Tween-80 from without Tween-80.
- Overall this is an interesting paper but the title, the introduction and the results and discussion are too general for this journal. I'd suggest the Authors to divide the result from the discussion and to rewrite these two paragraphs in order to make it more scientifically sound and more appropriate for a publication in IJMS.
Response: Thanks for your suggestion; your suggestion is incorporated. The results and discussion are two different sections and were mistakenly written its title together. In the revised manuscript we corrected it.
- The conclusions also need to be more specific and needs to relate to the crop used for the study.
Response: Thanks for your suggestion; your suggestion is followed. We modified the conclusion.
Reviewer 2 Report
1. Title. Please think about the title. The title should be short (manuscript preparation) and communicate the processes and purpose of the research undertaken. It should make the reader curious with a certain mysteriousness. In my opinion, the title should suggest the possibility of a phenomenon that will be explained in the manuscript rather than informing in advance about the research results. I leave the decision to change the content of the title of the manuscript to the authors.
2. Purpose of the paper. The purpose of the study, which is contained in the last sentence of the Introduction chapter, is too general. It does not answer the questions that should be included in the purpose of scientific papers: why and how?
3. The Material and Methods chapter does not fully justify to the reader some of the studies undertaken: e.g., why were yeast fermentation and bioethanol assai and the mechanism of biochar and nitrogen application improvements on biomass digestibility studies done? There is a lack of justification in this chapter for precisely choosing the agent Tween-80 for testing and how it was used—no information on why this agent just was accepted for research.
4. Because, in the manuscript, there is a chapter entitled Results and Discussion, and the next chapter is Discussion, I suggest you consider changing the name of this chapter to Summary.
5. Chapter Conclusions. First sentence: "Based on our results, we hypothesized that biochar in combination with N fertilizer had a significant impact on biomass digestibility". According to academic knowledge, a hypothesis is put forward before the research, and an attempt is made to justify or reject it. It is unprofessional to make a hypothesis in Conclusions, which should be included in the objective of the research work described. In addition, the content in this chapter is very sparse. I believe that the sentence used as the title of this manuscript is more eligible to be placed here than where it is.
Apart from these comments, which I leave to the authors to decide whether to include, I find this manuscript interesting from a scientific point of view.
Author Response
Point 1: Title. Please think about the title. The title should be short (manuscript preparation) and communicate the processes and purpose of the research undertaken. It should make the reader curious with a certain mysteriousness. In my opinion, the title should suggest the possibility of a phenomenon that will be explained in the manuscript rather than informing in advance about the research results. I leave the decision to change the content of the title of the manuscript to the authors.
Response 1: Thanks for your suggestion; your suggestion is followed and the title is improved. The new title of the manuscript is “Effects of Biochar and Nitrogen Application on Rice Biomass Saccharification, Bioethanol Yield and Cell Wall Polymers Features”
Point 2: Purpose of the paper. The purpose of the study, which is contained in the last sentence of the Introduction chapter, is too general. It does not answer the questions that should be included in the purpose of scientific papers: why and how?
Response 2: Thanks for your suggestion; your suggestion is followed and added the detail at the end of the introduction.
“Thus regarding the importance of biomass composition, it is important to determine the impacts of different biochar rates in combination of N fertilizer on cell wall features and cellulose monomers of rice crop.
In the present study, we collected the rice crop biomass samples from the experimental field treated with different biochar and N fertilizers. Then, we determined major alteration of cell wall compositions and wall polymers features in the mature straw of rice crops. Based on comparative and correlative analyses, this study identified the main factors of plant cell walls that predominately determine biomass enzymatic saccharification and bioethanol production in rice crop under different treatments.”
Point 3: The Material and Methods chapter does not fully justify to the reader some of the studies undertaken: e.g., why were yeast fermentation and bioethanol assai and the mechanism of biochar and nitrogen application improvements on biomass digestibility studies done? There is a lack of justification in this chapter for precisely choosing the agent Tween-80 for testing and how it was used—no information on why this agent just was accepted for research.
Response 3: Thanks for your suggestion; your suggestion is as followed. “1% tween-80 was added to determine its impact on biomass saccharification in rice crop” added in the material and methods section 4.3.1. And we added in the third paragraph of the discussion why this agent just was accepted for research.
“Tween-80 is a non-ionic surfactant that has been widely used to enhance the enzymatic digestibility of pretreated biomass having high stability, good compatibility, and high solubility in the aqueous enzymatic hydrolysis system [42, 4].
Point 4: Because, in the manuscript, there is a chapter entitled Results and Discussion, and the next chapter is Discussion, I suggest you consider changing the name of this chapter to Summary.
Response 4: Thank you for your valuable suggestion. This was typing mistake. There are two different chapters Results and chapter 4 is discussion. We corrected it.
Point 5: Chapter Conclusions. First sentence: "Based on our results, we hypothesized that biochar in combination with N fertilizer had a significant impact on biomass digestibility". According to academic knowledge, a hypothesis is put forward before the research, and an attempt is made to justify or reject it. It is unprofessional to make a hypothesis in Conclusions, which should be included in the objective of the research work described. In addition, the content in this chapter is very sparse. I believe that the sentence used as the title of this manuscript is more eligible to be placed here than where it is.
Response 5: Thanks for your valuable suggestion. We modified all the conclusion according to your suggestion.

Reviewer 3 Report
The paper submitted to the International Journal of Molecular Sciences deals with an interesting topic on how the application of biochar and N could modulate positively the plant biomass and the saccharification and bioethanol yield. The authors conducted a couple of experiments, where the experimental design sounds. One of the main results found by the authors was that both cellulose DP, Crl, and Xyl/Araratio were negatively correlated with Ara, dry matter, and biomass digestibility. However, the molecular aspects are completely missing and this is a major drawback since we are dealing with a molecular Journal. The Results and especially section is very descriptive, it is not enough to report whether your results are in line or not with previously published data but why? This is not possible with the actual dataset. Therefore I cannot recommend this paper for publication in such a prestigious Journal.
Author Response
The paper submitted to the International Journal of Molecular Sciences deals with an interesting topic on how the application of biochar and N could modulate positively the plant biomass and the saccharification and bioethanol yield. The authors conducted a couple of experiments, where the experimental design sounds. One of the main results found by the authors was that both cellulose DP, Crl, and Xyl/Araratio were negatively correlated with Ara, dry matter, and biomass digestibility. However, the molecular aspects are completely missing and this is a major drawback since we are dealing with a molecular Journal. The Results and especially section is very descriptive, it is not enough to report whether your results are in line or not with previously published data but why? This is not possible with the actual dataset. Therefore I cannot recommend this paper for publication in such a prestigious Journal.
Response: Thank you so much for reviewing our manuscript. The manuscript evaluated biomass digestibility, cell wall feature and hemicelluloses monomers thus its falls fully within the scope of the special issue “Polymers from Renewable Resources 2.0” of IJMS. As biochar coupled with nitrogen fertilzer changed the cellulose and hemicellulose content. We further planned to perform an extensive study, to confirm the expression level of GT43, GT61 genes of hemicellulose, CECA gene of cellulose and 4CL, C4H, CCR, COMT, FSH, PAL gene of lignin. We further intend to perform 2DHSQC NMR of lignin monomers, side chain linkages, pCA, FA content to know the effect of biochar coupled with nitrogen. Therefore the current reports fulfill the requirements for the special issue and for the journal scope.
Round 2
Reviewer 1 Report
The paper is greatly improved and can now be accepted for publication.
Author Response
Point 1: The paper is greatly improved and can now be accepted for publication.
Response 1: Thank you so much for accepting our manuscript for publication.

Reviewer 3 Report
Dear authors
Thank you for your reply but I still believe that the manuscript does not meet the high standard of the Journal and I will keep the final decision to the Handling Editor.
Bests
Author Response
Point 1: Thank you for your reply but I still believe that the manuscript does not meet the high standard of the Journal and I will keep the final decision to the Handling Editor.
Response 1: Your rejection is baseless, without any comment or suggestion its scientifically not correct to reject a manuscript. A reviewer's duty is to present the serious flaws in the manuscript and suggest comments. If an editor sent the manuscript for review, it means the editor already read the manuscript and it meets the standard of the journal and its special issue, a reviewer doesn’t have the right to reject a manuscript without revising it on the basis of scope.
